# Conserved and Diverse Traits of Adhesion Devices from *Siphoviridae* Recognizing Proteinaceous or Saccharidic Receptors

**DOI:** 10.3390/v12050512

**Published:** 2020-05-06

**Authors:** Adeline Goulet, Silvia Spinelli, Jennifer Mahony, Christian Cambillau

**Affiliations:** 1Architecture et Fonction des Macromolécules Biologiques, Aix-Marseille Université, Campus de Luminy, 13288 Marseille, France; silvia.spinelli@afmb.univ-mrs.fr; 2Architecture et Fonction des Macromolécules Biologiques, Centre National de la Recherche Scientifique (CNRS), Campus de Luminy, 13288 Marseille, France; 3School of Microbiology, University College Cork, Cork T12 YN60, Ireland; J.Mahony@ucc.ie; 4APC Microbiome Ireland, University College Cork, Cork T12 YN60, Ireland

**Keywords:** Bacteriophage, *Siphoviridae*, baseplate, receptor-binding protein, phage–host interactions

## Abstract

Bacteriophages can play beneficial roles in phage therapy and destruction of food pathogens. Conversely, they play negative roles as they infect bacteria involved in fermentation, resulting in serious industrial losses. *Siphoviridae* phages possess a long non-contractile tail and use a mechanism of infection whose first step is host recognition and binding. They have evolved adhesion devices at their tails’ distal end, tuned to recognize specific proteinaceous or saccharidic receptors on the host’s surface that span a large spectrum of shapes. In this review, we aimed to identify common patterns beyond this apparent diversity. To this end, we analyzed siphophage tail tips or baseplates, evaluating their known structures, where available, and uncovering patterns with bioinformatics tools when they were not. It was thereby identified that a triad formed by three proteins in complex, i.e., the tape measure protein (TMP), the distal tail protein (Dit), and the tail-associated lysozyme (Tal), is conserved in all phages. This common scaffold may harbor various functional extensions internally while it also serves as a platform for plug-in ancillary or receptor-binding proteins (RBPs). Finally, a group of siphophage baseplates involved in saccharidic receptor recognition exhibits an activation mechanism reminiscent of that observed in *Myoviridae*.

## 1. Introduction

Viruses that infect bacteria, commonly named bacteriophages or phages, play an upmost role in biogeochemistry, biomes, health (phage therapy), and industry. In this latter case, phages may cause fermentation disruptions and product inconsistencies that are among the primary causes of economic losses in the associated dairy fermentation industry. Conversely, phages can be added to foods to prevent *Salmonella* [1] or *Listeria* [2] contaminations.

The genome of phages consists of double- or single-stranded DNA or RNA. Phages exhibit a large spectrum of structural morphology: capsid-only, tailed, filamentous, etc. [3]. However, tailed phage (*Caudovirales*), phages with a tail attached to the capsid and a double-stranded DNA genome, dominate among those found in public databases, e.g., NCBI, with 58% of all phages possessing this morphology [3]. The *Caudovirales* order may be divided into five families, the three dominant families being the *Myoviridae,* phages with a contractile tail; the *Podoviridae*, phages with a short tail; and the *Siphoviridae*, phages with a long non-contractile tail (Figure 1). *Siphoviridae* are the most abundant tailed phages in the public databases (66%) followed by *Myoviridae* (20%) and *Podoviridae* (14%). In contrast, the structural depiction of phages collected in oceans worldwide reveals the dominance of tailless phages (79%), where tailed phages account only for 21% of the total. Among them, *Myoviridae* dominates (67%), followed by *Podoviridae* (28%) and *Siphoviridae* (5%) [3]. In the Mediterranean Sea, however, an NCBI-like distribution is observed with a dominant population of *Caudovirales,* in particular *Siphoviridae*. It should not be forgotten that the majority of the phages collected in meta-studies (oceans, seas, gut, soil) remain unclassified. Furthermore, the population of phages annotated in the public databases may not reflect their sheer diversity but the fact that they interfere, negatively or positively, with human activity. Phages that infect lactic acid bacteria (LAB phages) involved in fermentation processes are over-represented, a clear illustration of this bias [4]. Another form of bias in phage studies arose from the initial scientific interest on phages infecting *Escherichia coli*, including the model coliphages T4 [5,6], lambda [7], HK97 [8], and T5 [9], as many strains of this bacterium (e.g., K12) are harmless and very easy to cultivate. 

Siphophages feature a long, non-contractile, and flexible tail connected to an icosahedral or prolate capsid with the latter containing its dsDNA genome (Figure 1). During the last decade, there has been an upsurge of interest in siphophages, especially those infecting Gram-positive bacteria, driven by the public safety and economic relevance of their hosts, which started to slowly dissect their infection mechanisms [10,11,12,13,14,15]. Despite this increased interest, considerable knowledge gaps remain regarding the molecular details of the mechanisms by which they bind, puncture, and hijack their hosts. Any successful phage infection starts with the recognition of the host cell surface by specialized adhesion devices. Siphophages use a multi-protein recognition device at the tail distal end, referred to as the tail spike or the baseplate, to bind either to proteinaceous receptors or to carbohydrates (and associated motifs) located at the surface of the host cell wall [16,17,18] (Figure 1). The composition and architecture of phage adhesion machinery can vary from one group of phages to another, tightly associated with different host recognition mechanisms. Here, we review the features of siphophage adhesion devices centered around their structural building blocks, the tape measure protein (TMP), the distal tail protein (Dit), and the tail-associated lysozyme (Tal), examining their common features and their differences. We pursue by inserting in this frame the different peripheral proteins such as the receptor-binding proteins (RBPs), the tail fibers, and other ancillary proteins. 

## 2. Modules Forming the *Siphoviridae* Tail Tip

### 2.1. The TMP-Dit-Tal Triad

The overall organization of the morphogenesis modules of most phages is syntenous and follows the lambda model with packaging, capsid, and tail morphogenesis functions encoded. Within the latter are the *tmp*, *dit,* and *tal* triad, and in most siphophages examined to date, the *tmp* gene is (immediately) followed by the *dit* and *tal* genes (Figure 2a). This sequence is roughly mirrored by the gene products topology in the virion tail tip (Figure 2b). The Dit hexamer interacts with the Tal trimer N-terminal domains. The TMP, which determines the virion’s tail length [19], carries its main course in the lumen of the stacked Major Tail Protein (MTP) rings and Dit hexamers. Its C-terminus abuts the Tal central cavity and interacts strongly with it [15,19]. Altogether, TMP, Dit, and Tal participate in the first assembly module of the tail, the Initiator Complex (IC) [20,21]. This IC complex is then completed by the RBP and other baseplate proteins, if present, as well as with the MTPs and the tail terminator, the protein that blocks the tail extension, resulting in the complete tail that is ready to attach to the capsid connector [20,22]. 

Dits of known structures are composed of two domains [15,24,29,30,31]. The N-terminal domains form a ring with two layers of β-strands (Figure 2c,d). This "ring domain" shows a split barrel-like fold similar to that found in phages’ MTP [32] and Hcp, a Type-VI secretion system (T6SS) protein [33,34,35]. A long kinked extension (the “belt”) of four β-strands embraces the next Dit core in the hexameric ring. This ring delineates a 40Å-wide channel to allow the transit of the dsDNA genome during infection. Dits from phages SPP1, p2, TP901-1, T5, and 80α possess C-terminal domains located at the ring periphery, which do not contact each other [15,24,29,30,31]. The first four display a galectin fold supplemented in phage p2 by a long extension (the “arm”, residues 147-188) having a critical role in baseplate assembly via the formation of a three-digit hand that anchors the N-terminal domain of the RBP (ORF18, see below) [24]. In phage 80α, a shorter loop is also involved in RBP binding, and a second loop establishes contacts with the MTP. In contrast, the fold of phage T5 C-terminal domain is that of an OB domain [31]. 

Tals of known structures include those of siphophages p2 [24], A118 (PDB ID: 3GS9), MuSo2 (PDB ID: 3CDD), and 80α [15] (Figure 2c,d). Furthermore, their fold is similar to those of gp27 of T4 myophages [5] and of the type VI secretion system (T6SS) spike protein VgrG [34,36]. Tal of phage p2 is 398 residues-long protein and harbors four domains [24]. It is found in closed or open conformations depending upon the state of the baseplate (see Section 4). Tals from most phages, however, are longer than ~400 residues, and in some cases reaching up to 2000 residues or more [28]. HHpred [37] analysis of these Tals reveals that they possess an N-terminal domain resembling the Tal domains mentioned above, for which the structures have been determined, followed by an extension exhibiting large structural and functional diversity. In phages lambda [7], T5 [38], and SPP1 [39], this extension harbors or binds the receptor-binding domain (RBD) that attaches to the LamB, FhuA, or YueB receptors, respectively. In phages TP901-1 and Tuc2009 [40], as well as in staphylococcal phage 80α [15], this extension possesses a lysin domain (hence the original name of Tal) that cleaves the peptidoglycan upon infection. Structure prediction of phage 80α Tal C-terminus suggests that it may also possess a peptidoglycan cleavage activity [15]. In some streptococcal phages as DT1 [41] or STP1 [28], Tals carry one or several carbohydrate-binding domains, capable of binding the cell wall surface saccharidic receptors. However, many Tal extensions are not annotated functionally. 

Little is known on the TMPs structure. Functionally, it has been shown by using partial deletion mutants that they control the tail length in a quasi-linear fashion [19,42]. Parts of the TMP, however, are not involved in tail length determination but are believed to interact with the tail terminator with their N-terminus and the Dit and Tal proteins with their C-terminus [19]. In some cases, they also incorporate an unfolded lysin domain that becomes active upon TMP release from the tail [43]. Structurally, they are predicted as helical, with transmembrane predicted helices (TMH) at their center [19]. In Gram- bacteria, it has been suggested that TMP interacts with proteins from the inner membrane (IM) in which its TMHs form a pore when it is ejected from the tail, thereby allowing an easier transfer of dsDNA into the cytoplasm [8]. The only experimentally determined TMP (partial) structure from phage 80α reveals that it assembles as a trimer [15], a TMP monomer per Tal monomer, and confirms that the TMP C-terminus interacts strongly with the Dit and Tal proteins, in agreement with the formation of a defined and strong initiation complex [20].

In addition to the above core architecture of Dit and Tal, a large diversity of decorating modules has been observed. Dits can harbor one to two carbohydrate-binding modules (CBMs) inserted in the loops of their peripheral galectin domains (Figure 2c). These functionalized Dits, termed "evolved Dits" [44], have been documented in *Lactobacillus* phages [44,45], lactococcal phages [46,47], streptococcal phages [28], and beyond [46]. Of note, these CBMs are often observed to share the BppA CBM fold [46]. Evolved Dits may be present in up to 30% of all phages from the lactococcal 936 group and 100% of the dairy streptococcal phages from the dominant *cos* and *pac* groups [28]. Furthermore, using fluorescence microscopy, they have been shown to bind to the same host as the *bonafide* RBP, although with a lower affinity [46]. Besides the large diversity of their C-terminal extensions, Tals may also be “evolved Tals” possessing CBMs, although less frequently than Dits. These evolved Tals harbor two decorating CBMs in their gp27-like N-terminal structural domain, as exemplified in streptococcal phages of the *pac* group [28] and beyond [48]. Other CBM decorations have also been mentioned in MTPs from phage lambda [49], phage SPP1 [50], and lactococcal phages [47]. In the latter case, the functionality of the CBMs has been demonstrated [47].

### 2.2. RBPs and Ancillary Proteins

While the annotation of the TMP-Dit-Tal triad is relatively straightforward, annotation of the remainder of the baseplate components is more challenging. An important observation, however, is that all baseplate-encoding genes (with few exceptions) are located in the genome between the *tmp* and the *holin/lysin* genes. This includes the Dit-Tal diad, the RBP, and possibly baseplate ancillary proteins. A source of complexity is that in most phage genomes, additional genes, which may or may not be functionally annotated, are also located between these boundaries. Therefore, one has to evaluate which gene products are candidate baseplate components. This involves widely used experimental methods such as SDS-PAGE and mass spectrometry, mutational analysis [20], as well as topological and structural comparison with other phages using HHpred [37]. 

Atomic structures of the RBPs of a small number of siphophages are available, i.e., those from lactococcal phages p2 [51], bIL170 [52], TP901-1 [53], Tuc2009 [54], and 1358 [55,56], as well as from the *Listeria* phage PSA [13] and staphylococcal phages ϕi11 [57] and 80α [15], all assembled as homotrimers. The lactococcal and listerial phage RBPs share topological elements (Figure 3). All assemble distinct domains permitting to bind to the rest of the baseplate (N-terminus), the receptor-binding domain (RBD; C-terminus), and a linker between both. In phage p2, the RBP N-terminus "shoulder" domain possesses an immunoglobulin fold. It is followed by an interlaced trimeric β-helix (the "neck") that maintains the RBP (termed the "head") (Figure 3b) [51]. All the lactococcal and listerial RBDs share topological elements such as a core β-barrel of 7-9 β-strands, with extra loops and strands in Tuc2009 and 1358 RBPs (Figure 3a–e). Of note, these folds also share similarity with the Reovirus attachment protein s1 (PDB ID 1KKE) and the head domain of the Adenovirus fiber (PDB ID 1QHV), present as a double Greek-key motif in their topology but without any sequence similarity [58,59,60] as well as with the phage T4 gp12 protein [61]. Conversely, the larger RBPs from phages ϕ11 and 80α exhibit a more complex topology with only very localized similarity compared to the other RBPs (Figure 3f,g) [15,62]. 

The modules that bind the RBP to the rest of the baseplate belong to one of two different types. These domains (residues 1 to 140) are very similar in phages p2 and 1358, assembling two 4-stranded anti-parallel β-sheets and a long helix contributed by each of the three domains allowing them to associate compactly (Figure 3a,b) [51,55]. It has been proposed that the same type should be also present in phage bIL170, which is highly related to p2. Strikingly, this domain has been uncovered in the Type VI secretion system (T6SS) trimeric protein TssK that ensures the binding of the tail to the membrane complex (see Section 4) [63,64]. Conversely, in lactococcal phages TP901-1 [30], Tuc2009 [54], listerial phage PSA [13], and staphylococcal phage 80α [15], the insertion of the RBP in the baseplate involves an N-terminal three helix bundle and loops that dock in cavities provided by the rest of the baseplate (Figure 3c–g). In listerial phage PSA, the RBP results from the fusion of a RBP module and a BppU module (as those of phage TP901-1) that binds to the rest of the baseplate [13]. Finally, in many phages, the baseplate incorporates ancillary proteins in order to maintain the RBPs or to provide additional binding domains. More details on these proteins are provided in the sections below.

## 3. Architecture of Siphophage Adhesion Devices Targeting Proteinaceous Receptors

Phages have adapted to bind to cell surface proteins or cell wall polysaccharides on the host cell, or in some cases, both. Phages that bind proteins are widespread among Gram-negative bacteria; their outer membrane (OM) is rich in membrane-embedded proteins that phages use to hijack their host. These phages historically represent those studied in most detail such as those infecting *E. coli*: lambda, HK97, and T5. Conversely, only a few examples of phage infecting Gram-positive bacteria, including the *Bacillus subtilis* phage SPP1 and the c2-group *Lactococcus lactis* phage c2, bind to a host membrane protein [16].

### 3.1. Gram-Negative Host

#### 3.1.1. Phage lambda

The genome of phage lambda has been thoroughly analyzed, as reviewed by Casjens and Hendrix [7]. However, structural knowledge on its host adhesion device is still fragmented. The gpH gene product has been assigned the TMP (Figure 2a). Downstream of gpH, gpM HHpred analysis reports a strong hit with phage T5 Dit (Table 1), suggesting that its Dit is only composed of the ring domain, and is devoid of any extension, an OB-fold in T5 and galectin in other phages (Figure 2a). The three following ORFs, gpL, gpK, and gpI, have been reported to encode tail-associated components (Figure 2a, Figure 4a). The gpL protein contains two domains, the C-terminal one coordinating an iron-sulfur 4Fe4S cluster [65], while the N-terminus did not return any HHpred hit. Although the function of this cluster is unknown, it may play a role in stabilizing the conformation of gpL within the tail similarly as iron ions stabilize the membrane-piercing proteins of the enterobacteria myophages P2 and phi92 [66]. In contrast, gpK is assigned a peptidase domain by HHpred (99% probability; residues 52–182 out of 199 in total). Strikingly, TMHMM analysis [67] of gpI reports the presence of two transmembrane helices (residues 68-91 and 96-118) in this 190 residues protein, and HHpred identifies a α−β−α−β motif for its first 64 residues. The ensuing gpJ protein comprises 1132 residues. Three copies of gpJ form the central fiber at the tail tip that binds to the host protein receptor. The 149 C-terminal residues of gpJ interact with the *E. coli* receptor LamB, a membrane porin involved in maltose uptake [7]. HHpred analysis of gpJ reports hits with the N-terminus of phage Tal proteins (Table 1). However, this segment is not located at the N-terminus of gpJ but starts at residues 326-330. If we assume that it resembles the classical Tal N-terminus domain of other phages, this domain should cover gpJ residues ~185-520. Another hit reports fibronectin III repeats (Table 1) that may serve as linkers between the N-terminal structural domain and the C-terminal receptor-binding domain (Figure 4a). The two following ORFs, encoded by the side tail fiber gene (*stf*) and the tail fiber assembly gene (*tfa*), comprise the lambda long and thin side tail fibers. Of note, the version of lambda virtually used in all laboratories and referred to as the wide-type lambda, contains a frame mutation in the *stf* gene and is devoid of side tail fibers [68]. Although these side tail fibers are not essential, they can reversibly bind to another host membrane protein OmpC, thereby assisting the virions to encounter their host for infection [69]. HHpred analysis indicates that the C-terminal domains of Stf resemble the tip domains of the long tail fiber protein gp37 from the myophage T4 (95.2% probability, residues 578-774 out of 774 in total) exhibiting an α/β collar domain, an elongated six-stranded antiparallel β-strand needle domain, and the OmpC-binding domain made up of β-strands and long loops [70] (Figure 4a). Therefore, the long tail fibers of the myophage T4 and siphophage lambda, both bind the *E. coli* OmpC receptor and can be functionally substituted from one phage to the other [71], likely share a common molecular architecture. Regarding Tfa, which assists the proper assembly of the Stf trimer and is a component of mature virions [68], it likely adopts a fold similar to that of the myophage Mu Tfa protein as suggested by HHpred analysis (99.4% probability, residues 62-192 out of 194 in total), except for the first 61 residues (Figure 4a). Indeed, the N-terminal domain of the Mu Tfa protein associates with the Mu Stf receptor-binding domain that binds to lipopolysaccharides and is structurally different from the T4 and lambda tail fiber proteins [72].

#### 3.1.2. Phage T5

The genomic organization and architecture of the coliphages T5 and lambda differ in many aspects. The T5 protein pb2 represents the TMP. While the TMP was first thought to be the major component of the central tail fiber, it is not long enough to span the whole tail tube and tail tip, and it could not be detected in the tail tip [9,77]. Instead, the pb9, pb3, pb4, and pb5 proteins assemble to form the T5 tail tip. Downstream of pb2, pb9, and pb3 are the Dit and Tal proteins, respectively, which together form a conical structure (Figure 2a, Figure 4b). The Dit, which contains a C-terminal extension as compared with the Dit of lambda (Figure 4b), forms the hexameric ring of the upper part of this siphophage tail tip [31]. Similar to the lambda gpJ protein, HHpred analysis of pb3 reports hits with the N-terminus of phage Tal proteins and with fibronectin III repeats (Table 1). As in phage lambda, this segment is not located at the N-terminus of pb3 but starts at residue ~530. If we assume that it resembles the classical Tal N-terminus domain of other phages, this domain should cover pb3 residues 386-754 (Table 1, Figure 4b). The T5 central fiber is formed by pb4, likely as a trimer, and maybe also by the C-terminus of pb3. HHpred analysis indicates that the N-terminal domain of pb4 folds into fibronectin III repeats (97.4% probability, residues 6-319 out of 685 in total) but does not return any hit for its C-terminal domain (Figure 4b). Lastly, the pb5 protein (585 residues), located at the extremity of the central tail fiber, is the monomeric receptor-binding protein that binds to the *E. coli* ferrichrome receptor FhuA. Pb5 exhibits an elongated shape in solution and does not resemble any protein of known structure [78]. Surprisingly, the *pb5* gene does not belong to the tail assembly operon in which the *pb4* gene is followed by the *p132* and *pb1* genes encoding for the Long Tail fibers (LTF, pb1) and their collar proteins (p132) (Figure 2a) [9]. The LTFs are formed by trimers of pb1, anchored to the hexameric Dit ring via p132, that recognize polymannose O-antigens at the host cell surface. The pb1 C-terminal receptor-binding domain (residues 970-1263 out of 1396 in total, the last 133 residues being the intra-molecular chaperone that is auto-proteolyzed after folding) is formed by intertwined β-sheets with a bullet-like shape, which is different from the predicted fold of the lambda LTF C-terminal domains (Figure 4b) [73]. HHpred analysis of p132 reveals that this protein possesses the TP901-1 BppU N-terminal domain fold (98.8% probability, residues 10-138 out of 140 in total) (Figure 4b). Multiple copies of BppU in the phage TP901-1 baseplate interact with the Dit ring through their N-terminal domain [30]. Therefore, the fold of proteins that anchor receptor-binding proteins to the Dit central platform appears to be conserved among siphophages. However, the site(s) of association between p132 and pb1 to attach LTFs to the T5 host recognition device remains to be determined.

### 3.2. Gram-Positive Host

#### 3.2.1. Phage SPP1

Phages that bind Gram-positive bacteria using proteins as anchors are a minority. Few proteins are available at the bacterial surface, as Gram-positive bacteria are surrounded by a thick cell wall including peptidoglycan and other surface polysaccharides [79]. *B. subtilis* phage SPP1 is the best-studied representative siphophage that binds to proteinaceous receptor. The SPP1 tail tip establishes irreversible contacts with the host receptor membrane protein YueB after an initial and reversible interaction with glycosylated teichoic acids [80]. The SPP1 TMP is represented by gp18 [81] (Figure 2a). The gp19.1 Dit protein connects the tail tube to the long tail fiber protein, gp21 (Tal). The hexameric ring of gp19.1, showing C-terminal domains protruding out of the central core, associates with the trimeric gp21 (Tal) N-terminal domain [29,82] (Figure 4c). This Dit-Tal complex, which closes the tail tube in the SPP1 virion, opens upon host recognition and binding leaving an open channel for DNA ejection [82,83]. This closed-to-open conformational switch of Tal proteins, also observed in the *L. lactis* phage p2 [24], maybe a signal that is common to Gram-positive-infecting siphophages to trigger genome ejection. HHpred analysis of gp21 indicates that its first 404 residues fold as the A118 Tal N-terminal domain and are followed by a fibronectin repeat (Table 1, Figure 4c). The atomic structure of the SPP1 Tal C-terminal domain (residues 519-1111) remains unknown, although the Phyre protein fold recognition server [84] indicates that it may adopt a α-helical fold similar to that found in the structure of *Podoviridae* and *Myoviridae* tail fiber and tail spike proteins. The cryo-electron microscopy (cryoEM) observations of SPP1 virions mixed with the ectodomains of the YueB receptor show that, after the tail tip interaction with YueB and its loss, the Tal C-terminal domain adopts heterogeneous conformations [39]. Although four proteins, gp22, gp23, gp23.1, and gp24, are encoded between the Tal and lysis proteins (Figure 2a), their role and structure are poorly documented. In particular, the presence of these proteins in the tail spike remains to be determined. The gp22 crystal structure revealed a protein fold similar to the N-terminal “shoulder” domain of the lactococcal p2 RBP [74], indicating it may be a tail spike component (Figure 4c). The crystal structure of the gp23.1 hexameric assembly was also determined (Figure 4c), but it could not provide any conclusive insights into its function [75]. 

#### 3.2.2. Lactococcal Phages c2 and bIL67

Phages c2 and bIL67 belong to the c2 group of lactococcal phages that represent one of 10 groups that infect this bacterial species [85]. The c2 group contains the only lactococcal phages that are described to bind to a proteinaceous receptor, an orthologue of *B. subtilis* YueB, Pip in the case of c2, YjaE for bIL67. The annotation of the adhesion device of these two phages is scarce and confused [86]. A striking example is the lack of identification of the TMP. Several pieces of evidence suggest that protein l10 is the TMP: the *l10* gene is the longest of the whole genome; it is located in between the capsid/tail genes and the tail tip and holin genes; its product exhibits a predicted transmembrane helix; it facilitates DNA entry in the host by binding to the host membrane protein [23]. Two non-structural proteins are encoded by genes downstream of the putative TMP, the first being non-annotated (l11) and the second annotated as a terminase (l12) [86]. HHpred analysis of the following ORFs, l13-l16, encoded by genes located upstream of the holin (l17) does not provide conclusive evidence of their functions. ORF l13 is 100 amino-acids long, while ORF l14 is 638 residues long. To note, Millen and Romero showed that recombinant phages bIL67 in which the c2 ORFs l14-l15-l16 were exchanged with bIL67 ORFs 34-35-36 were able to bind Pip, the c2 receptor [23]. The sizes of l13 and l14, their position in the genome, and the fact that l14 is host-specific suggest that l14 would be the best candidate to bind to the Pip receptor and that l13-l14 may be the Dit-Tal diad (Figure 2a). Conversely, the following ORFs, l15 and l16, are annotated as carbohydrate-binding proteins that may be involved in the first, reversible, host recognition step.

## 4. Architecture of Siphophage Adhesion Devices Targeting Hosts’ Surface Polysaccharides

These phages are readily distinguishable from those that bind to a proteinaceous receptor even based on observations of low-resolution negative staining EM (nsEM) images. Whereas phages that bind to a host protein have an elongated tail tip, phages binding to cell wall surface polysaccharides possess a bulky distal tail feature named the baseplate. This large structure encompasses a high number of RBPs (typically 18 to 54) as avidity is required to overcome the rather low RBP affinity for sugars, as compared to high protein–protein affinity observed for example in phage T5 [78]. The highlighted examples below illustrate the conserved and diverse parts of these adhesion devices.

### 4.1. Binding-Ready Baseplate

#### 4.1.1. Lactococcal Phage TP901-1

The X-ray structure of a TP901-1 baseplate complex, comprising ORFs 46 (Dit), 48 (upper baseplate protein, BppU), and 49 (RBP), but without the Tal (ORF47), has been determined [30,87,88]. The TP901-1 baseplate (320Å wide and 160Å high, mass of 1.76 MDa) exhibits a 6-fold symmetry (Figure 5a). From the proximal to the distal end, it is formed by a Dit hexamer surrounded by 18 copies of BppU holding 18 trimers of the RBP (Figure 5a). The Dit protein possesses a structure similar to that of phages SPP1 [29] and p2 but without the arm extension of the latter [24]. The 18 BppU assemble as six asymmetric trimers connecting the Dit central core. Each monomer is composed of an N-terminal globular domain (BppU-Nt, 1-122) followed by a helical trimeric linker and a globular C-terminal domain (195-299) [30]. The C-terminal domains fold as ß-sandwiches and assemble as a 3-fold symmetric triangular-shaped trimer. This structure binds to the stem domains of three RBP trimers (Figure 5a). The conservation of the residues involved in the BppU/RBP interactions suggests that common architectural themes are found among lactococcal P335 group phages [30]. In the structure, the RBPs point in the opposite direction to the capsid, favorable to interacting with the host without further conformational change (Figure 5a).

#### 4.1.2. Phage Tuc2009

EM studies have demonstrated the structural resemblance of the P335 phage Tuc2009 baseplate with that of TP901-1 [89,90]. Although the overall sequence identity between Tuc2009 and TP901-1 phage genomes is higher than 95%, the RBPs differ significantly: the stem and neck domains display high sequence identity, while the head domain displays no identity at all. Indeed, both phages target different *L. lactis* strains (UC509.9 and 3107, respectively), which produce different host cell wall polysaccharides in terms of composition and structure [17,79]. The major difference between these two baseplates is the presence of an additional protein termed BppA in the Tuc2009 baseplate, whose gene is located between *bppU* and *bppL* (Figure 2). It was shown that this protein increases the binding specificity and/or affinity of the Tuc2009 tripod to its host receptor [90]. The structure of the Tuc2009 baseplate “tripod” assembles the BppU trimeric C-terminal domains to which are attached three RBP trimers and three BppA monomers [54]. As expected, the RBP head differs strongly from that of phage TP901-1 despite sharing a core with similar fold (Figure 3c,d). BppA assembles a compact 150 residues domain (domain 1), a junction segment, and a 50 residue β-sandwich (domain 2) harboring the extension that links it to a BppU C-terminal extension. BppA domain 1 exhibits similarity with carbohydrate-binding modules (PDB IDs 1GUI and 1GU3). BppA domain 2, together with the BppU extension, is very similar to a titin domain (PDB ID 2NZI). The BppA CBM was later found to be ubiquitous in decorations of lactococcal and streptococcal phages evolved Dits or MTPs [28,47,91]. 

#### 4.1.3. Listerial Phages A118 and PSA

The adhesion modules of listerial phages A118 and PSA have been thoroughly analyzed by informatics or experimentally [13,62]. HHpred analysis revealed that phage PSA exhibits a classical Dit, while its Tal protein possesses an extension of ~300 amino acids after the N-terminal structural domain, harboring a glycosidase domain at its extremity. By analogy with phage TP901-1, the ORF downstream Tal is believed to form a hexamer of trimers composed of N-terminal BppU-like domains, followed by a kinked triple helix and terminated by a domain whose structure has been determined experimentally: three helical segments complex a cation and the trimeric RBP domain is a double β-barrel similar but not identical to the RBP head of phage TP901-1 (Figure 3e,c) [13]. It has been suggested that phage A118 topology [62] resembles that of Phage PSA. Its Dit is also classical, but its Tal does not possess any extension as it is 341 amino-acids long instead of 705 in PSA. The RBP following Tal is also a composite of two domains and its fold is likely very similar to that of phage PSA.

#### 4.1.4. Staphylococcal Phage 80α

The recent structure of staphylococcal 80α has uncovered many interesting features [15]. Besides its classical Dit and the structural domain of Tal (followed by a small part of the extension), it reveals how the RBP stem attaches directly to the Dit ring. The structure of phage 80α RBP is almost identical to that of phage ϕ11 [57], despite displaying different kink angles in their stem (Figure 3f,g). Two different hexamers of lateral fiber trimers or dimers have been identified and are connected to the MTPs *via* rings of BppU N-terminal domains similar to those observed in TP901-1. Another exciting feature is the first structural identification of a TMP segment. A trimer of C-terminal residues 1135-1154 has been identified tightly packed inside the trimeric Tal cavity. This definitively confirms that the oligomerization of the TMP is directed by the trimeric Tal and not the hexameric Dit in the IC. Furthermore, it suggests that upon Tal opening, as observed in phages p2 [24] and SPP1 [82], the TMP release may trigger portal opening and DNA release [15].

### 4.2. Activatable Baseplates

#### 4.2.1. *Lactococcus lactis* Phage p2

The baseplate of phage p2 (Dit, Tal, RBP) (Figure 6), a member of the 936 group of lactococcal phages, was expressed in complex with llama nanobodies (VHHs) (Figure 6d) [24]. Its structure assembles a Dit hexamer attached to a Tal trimer, this diad being surrounded by six RBP trimers (Figure 6b–d). The Dit galectin domain exhibits a long insertion in the loop immediately after its first β-strand. This elongated stretch, termed the “arm”, abuts to a loop of triangular shape, named the “hand”. The hand is inserted between the N-terminal trimeric domains (the “shoulders”) of the RBP (see below). As the Dit is in contact with the Tal trimer, the 6 to 3 change in symmetry is possible since two of the four domains of the Tal have similar topology, and each interacts with a Dit monomer ring domain, as described by the Rossmann group for T4 gp27 interaction with the tail [5]. Whereas the Dit exhibits a large ~40 Å central channel, the Tal is closed. There are no contact points between the trimeric Tal and the RBPs and, therefore, the Dit hexamer is a central hub to which Tals and RBPs are attached (Figure 6c). Surprisingly, the expressed baseplate revealed an "inverted" conformation. One would expect the head domains of the RBPs to point “downwards”, i.e., in the direction of the host cell surface, opposite to the capsid direction, as observed in phage TP901-1. Instead, the RBPs were observed in a “heads-up” conformation, pointing toward the capsid, an orientation that is incompatible with optimal adhesion to the host. However, there was no doubt on the validity of the structure of the baseplates crystal structure as this conformation was found in the negative staining EM (nsEM) structure of the expressed baseplate as well as in the full virion nsEM structure (Figure 6a). In both cases, the X-ray structure fitted perfectly within the baseplate region of EM reconstructions [24] (Figure 6c). However, comparing the X-ray structure of the baseplate complex with that of the expressed baseplate by nsEM revealed that the VHHs binding to RBPs expelled the second ring of Dit, attached back-to-back to the first one (Figure 5c,d). Strikingly, this Dit-2 maintains the RBPs conformation in the "heads-up" conformation through a contact between the RBP heads and the Dit "hand" loop extension (Figure 5c). The resolution of the nsEM map, however, does not allow a description of the atomic details of this interaction. By the addition of Ca^2+^ or Sr^2+^ to the crystallization medium of the baseplate (in the absence of VHHs), a different baseplate conformation was obtained in which the RBPs turn by 200° in a direction opposite to the capsid, a conformation named the “activated form” by contrast to the previous “rest form” (Figure 5e,f). The Tal trimer was also affected, resulting in its opening and the formation of a channel of ~32 Å diameter, large enough for dsDNA passage. Due to the new conformation, the RBPs and Tal establish contacts that lock the RBPs in their “heads down” conformation. This represents an activation mechanism related to that described in *Myoviridae* such as phage T4 [6]. This adhesion mechanism is indeed very different from that observed in phages that are "ready to bind" such as TP901-1, PSA, and many other phages. However, as we show below, this activation mechanism is widespread in siphophages, although its functional utility remains undocumented.

#### 4.2.2. *Lactococcus lactis* Phage 1358

The nsEM structure of the lactococcal phage 1358 virion [93] has been determined [25], as well as the X-ray structure of its RBP [55,56] (Figure 7 and Figure 3a). The trimeric RBP assembles two domains: a “shoulders”domain very similar to that of phage p2, and a “head” domain with a core resembling that of phages p2 or TP901-1 but decorated by several extra strands and loops (Figure 3a). The RBP’s X-ray structure was fitted into the virion’s baseplate structure. Strikingly, an extension assigned to 1358′s Dit inserts within the cavity formed by the three shoulder domains, similarly as in phage p2 (Figure 7a,b). Opposite to the shoulder domains, the head domains are in contact, without the second Dit extension, but with a structure belonging to a long MTP extension (Figure 7b,c). Of note, a second minor conformation of the phage baseplate, in which the RBPs were rotated by ~180°, was observed to spontaneously occur in the virion. This suggests that the baseplate activation mechanism may be similar, in many respects, to that of phage p2 during phage 1358 adhesion. 

#### 4.2.3. Other Phages

To date, a handful of siphophage virions and RBPs structures are available. However, thanks to the predictive power of tools such as HHpred, analysis of the baseplate genes may make it possible to identify putative phages with a baseplate activation mechanism. Without presenting an exhaustive view of phages with activatable baseplates, we selected a small number of putative candidates in different bacterial species that we analyzed with HHpred. *Listeria* phage P35 gp15, gp16, and gp17 share the p2 Dit, Tal, and RBP fold with 100% probability over their full length. A similar pattern is observed with *Listeria* phage P40, but with a CBM insertion in its Dit, which is also observed in other lactococcal 936 group phages. RBPs from *Leuconostoc* phages CHA, PhiLN025, and Phi1-A4 exhibit fold similarity with that of p2 RBP with ~99% probability. Phage LC3, which belongs to the P335 group of lactococcal phages (along with TP901-1 and Tuc2009), exhibits baseplate characteristics of phage p2 [94]. As mentioned above, the attachment mode of the RBPs shoulders domains to the Dits’ extensions, is shared between phages p2 and 1358. Surprisingly, TssK, a component of the type VI secretion system, possesses a shoulders domain similar to that of phage p2′s RBP [63]. The cryoEM structure of T6SS baseplate-like complex revealed that this similarity extends to the binding mode of TssK to its partner, TssG. Two triangular stretches of residues belonging to TssG serve to attach two TssK proteins to the baseplate hub [64]. 

## 5. Perspectives

Modular shuffling within phage adhesion modules has indeed been observed and documented previously [34]. It is, however, very gratifying to note that, with a rather limited number of known structures in the siphophage sphere, our analysis retrieved so many hits. This suggests that through evolution, siphophages have exchanged a small number of efficient modules and dispensed them in different architectures, resembling a LEGO-like game assembling virions with common structural bricks. Striking examples of shared modules are indeed the Dit and Tal N-termini, but also the BppU N-terminal structural domain and the BppA CBM. Conversely and importantly, it may be efficient to concentrate structural efforts on «unknown» parts of phage proteins, as they may allow a cascade of new assignments. In particular, the adhesion devices of "classical coliphages" such as lambda, T5 in addition to other model phages such as SPP1 or c2 represent viral dark matter, from a structural viewpoint, that deserves attention. Undoubtedly, with the advent of atomic resolution cryoEM, new exciting features will be revealed.

## Figures and Tables

**Figure 1 viruses-12-00512-f001:**
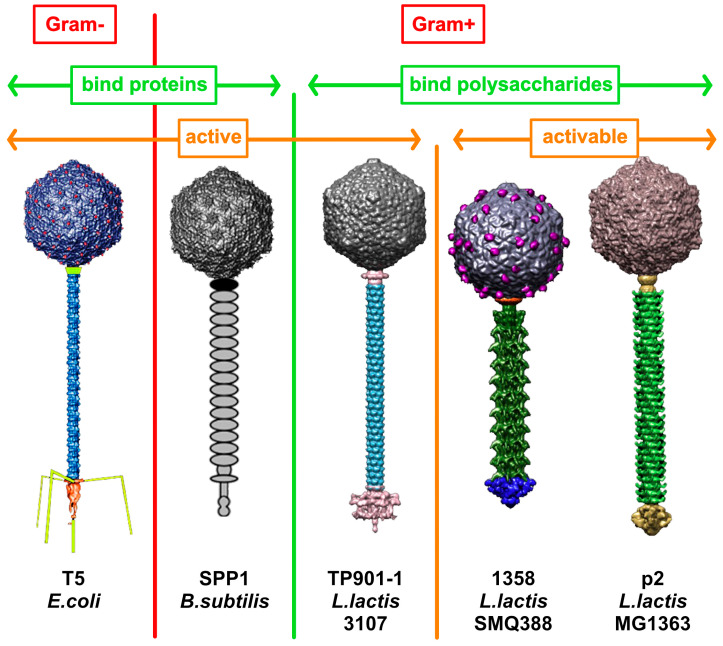
Examples of siphophages infecting Gram-positive and Gram-negative bacteria. Phages are classified between those binding to host-encoded proteins (green, left) or cell wall polysaccharides (green, right) and those with a ready to bind adhesion device (orange, left) and those with an activable adhesion device (orange right).

**Figure 2 viruses-12-00512-f002:**
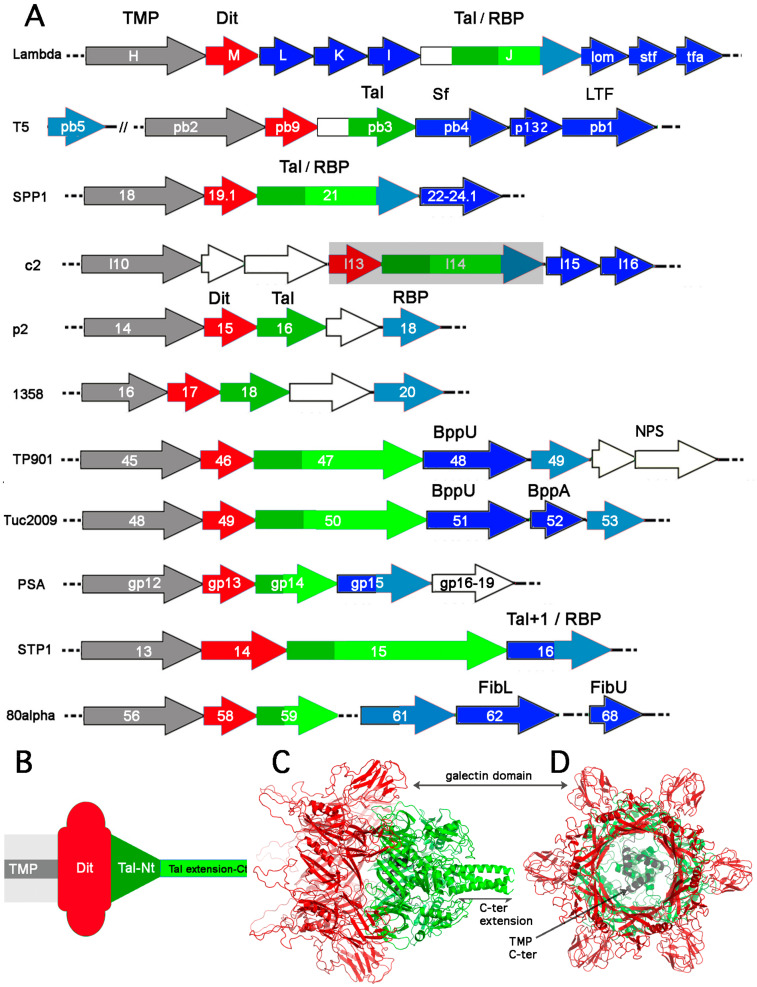
The adhesion device of siphophages. (**A**) Schematic representation of genes coding for the adhesion device of representative phages: lambda [7], T5 [9], SPP1 [9], c2 [23], p2 [24], 1358 [25], TP901-1 [21,26], Tuc2009 [27], PSA [13], STP1 [28], 80α [15]. Tape measure proteins (TMPs) are color-coded grey; distal tail proteins (Dits) are color-coded red; tail-associated lysozymes (Tals) are color-coded dark green (Nt-structural domain), light green (Tal extension), light blue for Tal extension tip bearing a receptor-binding domain; receptor-binding proteins (RBPs) are color-coded light blue; ancillary proteins and tail fibers are color-coded dark blue. White genes are non-structural. The c2 phage Dit and Tal are shaded grey as their annotation is highly hypothetic; (**B**) Schematic topology of the TMP-Dit-Tal triad (same color code as in (**A**)); (**C**) Example of a TMP-Dit-Tal triad from phage 80α baseplate [15] (same color code as in (**A**) and (**B**). The TMP is dark grey); (**D**) 90° rotation with respect to C.

**Figure 3 viruses-12-00512-f003:**
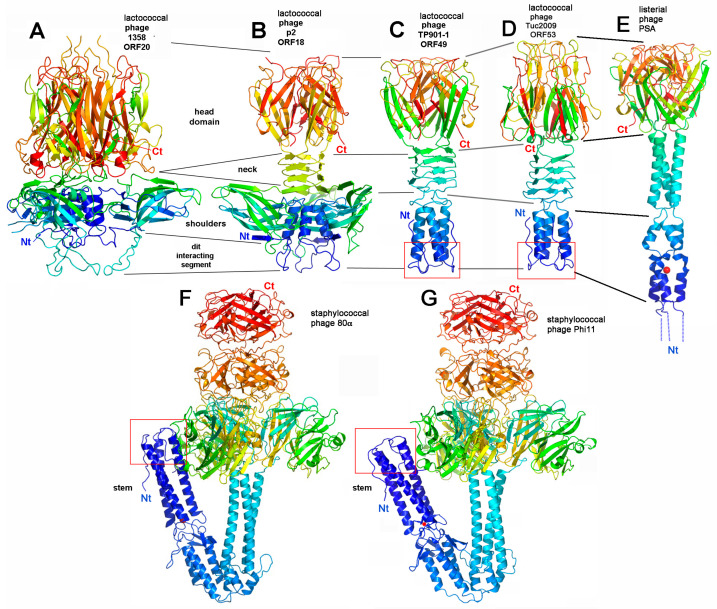
X-ray structures of RBPs from siphophages binding to host cell wall saccharidic receptors. (**A**) Lactococcal phage 1358 [55]; (**B**) Lactococcal phage p2 [51]; (**C**) Lactococcal phage TP901-1 [53]; (**D**) Lactococcal phage Tuc2009 [54]; (**E**) Receptor-binding domain and stem of listerial phage PSA [13]; (**F**) Staphylococcal phage 80α [15]; (**G**) Staphylococcal phage ϕ11 [57]. The various domains are identified: shoulders and stem, binding to the rest of the baseplate, the neck joining the shoulders/stem to the head. The head bearing the receptor-binding crevice. The stem N-terminus, involved in baseplate integration of the RBP, has been boxed in red.

**Figure 4 viruses-12-00512-f004:**
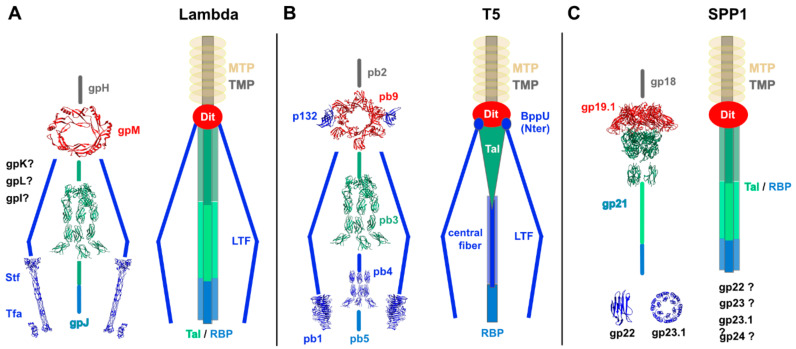
Architecture of the phage lambda, T5, and SPP1 host adhesion devices targeting membrane proteins. (**A**) *Right.* The lambda host adhesion device is formed by the Dit hexameric ring (gpM) and the long trimeric Tal / RBP (gpJ) central fiber. The positions of gpK, gpL, and gpI proteins in the lambda tail spike are not known. Ur-lambda also contains long tail fibers (LTFs) (Stf and Tfa). *Left.* Structural models, based on HHpred analyses, of gpM, gpJ middle domains (residues 326-449 and 570-811), Stf C-terminal domains (residues 578-774), and Tfa (residues 62-192). The Dit ring is a tilted view. The crystal structures used to produce these representations are PDB ID 4JMQ [31] for gpM, PDB IDs 3D37, and 5UTK for gpJ, PDB ID 2XGF for Stf [70], and PDB ID 5YVQ for Tfa [72]. (**B**) *Right.* The T5 host adhesion device is formed by the Dit hexameric ring (pb9), the trimeric Tal (pb3), and central fiber (pb4), and the RBP (pb5). LTFs (pb1) are anchored to the Dit ring via p132. *Left.* Ribbon representations of the pb9 (PDB ID 4JMQ) and pb1 (PDB ID 4UW8) [73] crystal structures. The Dit ring is a tilted view. Structural models, based on HHpred analyses, of pb3 (residues 528-709 and 735-849), pb4 (residues 6-319), and p132. The crystal structures used to produce these representations are PDB IDs 3D37 and 5UTK for pb3, PDB ID 5UTK for pb4, and PDB ID 3UH8 [30] for p132. (**C**) *Right.* The SPP1 host adhesion device is formed by the Dit hexameric ring (gp19.1) and the long trimeric Tal / RBP (gp21). The positions of gp22, gp23, gp23.1, and gp24 proteins in the SPP1 tail spike are not known. *Left.* Ribbon representations of the gp19.1 (PDB ID 2X8K), gp22 (PDB ID 2XC8), and gp23.1 (PDB ID 2XF7) crystal structures [74,75]. Structural models, based on HHpred analyses, of the gp21 N-terminal domains (residues 3-404 and 422-518). The crystal structure used to produce these representations are PDB IDs 3GS9 and 5E53 [76].

**Figure 5 viruses-12-00512-f005:**
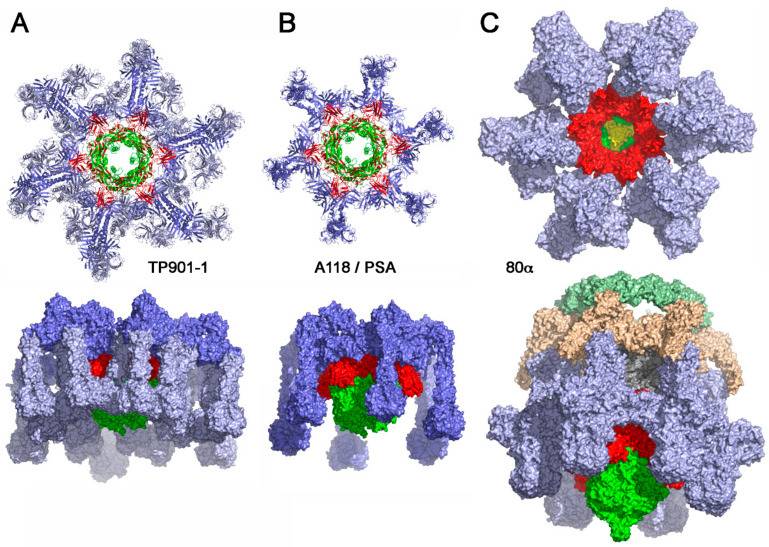
Structures of binding-ready baseplates from siphophages binding to a host cell wall saccharidic receptor. (**A**) X-ray structure of the baseplate from phage TP901-1; the Tal trimer has been modeled by addition of phage p2 Tal to the X-ray structure. (**B**) Model structure of the RBP from phages PSA or A118. (**C**) -cryoEM structure of phage 80α baseplate [15]. The color-coding is identical to that in Figure 2: Dit: red; tal: green: RBP: pale blue; ancillary proteins dark blue; in (**C**) Fiber proteins are colored beige and pale green (80α).

**Figure 6 viruses-12-00512-f006:**
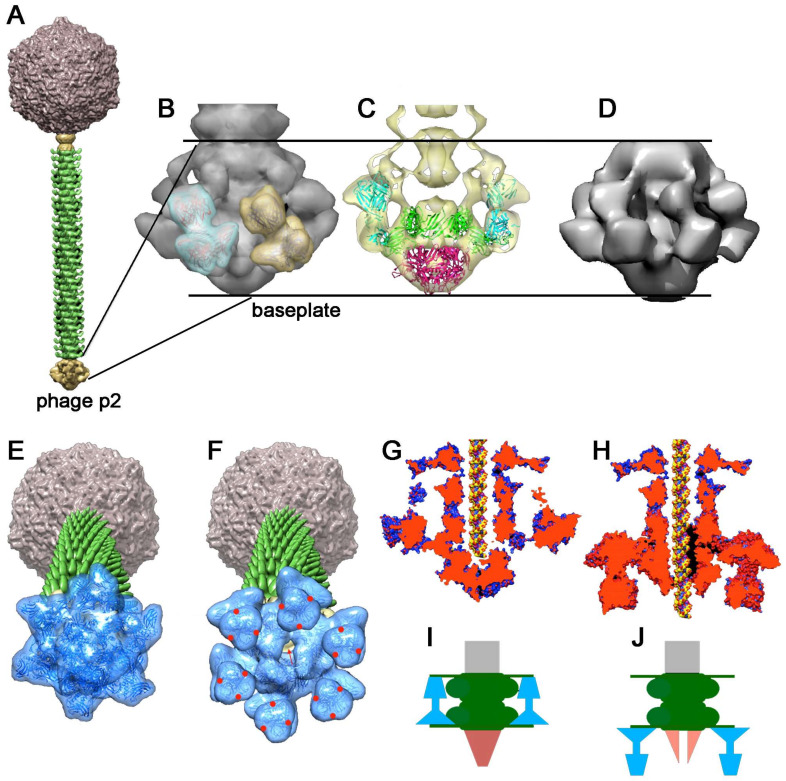
Structure of the activatable baseplates from siphophage p2. (**A**) negative staining EM (nsEM) structure of the phage p2 virion [92]. (**B**) Zoom on the baseplate with two RBPs fitted in the nsEM map. (**C**) Slicing of the nsEM map exhibiting the baseplate (Dits, Tals, and RBPs) fitted inside. (**D**) nsEM structure of the expressed baseplate; note the extra volume compared to the X-ray structure obtained in the presence of VHHs [24]. (**E**) Representation of phage p2 with the non-activated baseplate. (**F**) Representation of phage p2 with the X-ray determined activated baseplate localized at the tail extremity. The red arrows show the open Tal channel and the red dots the position of the receptor binding crevices. (**G**,**H**) sliced views of the structures shown in (**E**)- and (**F**)-, respectively. A model of dsDNA helix shown in the central closed or open channel. (**I**,**J**) schematic representation of the phenomenon associated with baseplate activation.

**Figure 7 viruses-12-00512-f007:**
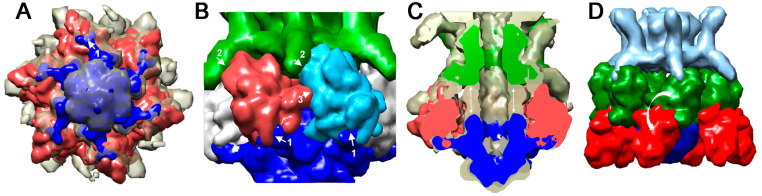
Structure of the activatable baseplates from siphophage 1358 [25]. (**A**) The nsEM structure of the non-activated baseplate viewed from the phage bottom. Tal and Dit are colored blue. The RBP is colored red. Note the "arm and hand" of Dits interacting with base of the RBPs (white arrow). (**B**) Lateral view of the baseplate showing the interactions of the Dit with the RBP N-terminal domain (shoulders) (1), the interaction of the MTP lateral extension with the RBP head domain (2) and lateral RBP contacts (3). (**C**) Sliced lateral view of the baseplate showing the non-assigned volume that can be attributed to a second Dit hexamer (white arrows). Tal and Dit are colored blue, RBP red, and MTPs green. (**D**) Lateral view of the nsEM maps of the baseplate in the non-activated state (green) and the activated state (red) resulting from a rotation of the RBPs attached to the Dits arms and hands.

**Table 1 viruses-12-00512-t001:** HHpred hits for the Dit and Tal proteins of lambda, T5, and SPP1.

Query	Target (PDB ID, Source)	Probability (%)	E-Value	Identity (%)	Query Residue Range (total)	Target Residue Range (Total)
lambda gpM (Dit)	6F2M, T5	94	0.42	11	4–77 (109)	12–85 (217)
lambda gpJ (Tal)	3D37, *N. meningitidis* MC58	95.9	0.18	14	327–499 (1132)	143–320 (381)
	3CDD, Shewanella MuSo2	95.8	0.18	10	326–499	146–324
	1WRU, phage Mu	94.1	0.69	16	326–499	140–318
	3GS9, *L. monocytogenes* A118	85.0	12	8	334–499	149–307
	5UTK, Receptor extracellular region (fibronectin III repeats)	98.7	1.6e-6	11	570–811	64–297
T5 pb3 (Tal)	3D37, *N. meningitidis* MC58	97.3	0.012	11	529–725 (949)	143–350 (381)
	3CDD, Shewanella MuSo2	97.1	0.011	12	528–709	146–339 (361)
	1WRU, phage Mu	96.8	0.026	15	529–716	141–339 (379)
	3GS9, Listeria	91.3	4.3	15	533–708	146–320 (342)
	5E53, adhesion molecule (fibronectin III repeats)	98.6	1.9e-5	13	690–949	133–295
SPP1 gp21 (Tal)	3GS9, *L. monocytogenes* A118	99.9	4.9e-25	12	3–404 (1111)	1–141 (342)
	3CDD, Shewanella MuSo2	98.9	4.1e-7	11	1–382	2–340
	3D37, *N. meningitidis* MC58	98.9	7.7e-7	10	1–381	2–336
	1WRU, phage Mu	98.9	9.2e-7	11	2–381	1–334
	5E53, adhesion molecule (fibronectin III repeats)	96.4	0.011	14	422–518	204–301

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
