# Peer review of "Conserved and Diverse Traits of Adhesion Devices from Siphoviridae Recognizing Proteinaceous or Saccharidic Receptors"

_viruses, 2020, doi:10.3390/v12050512_

Round 1

Reviewer 1 Report

Review of Viruses-789412 Cambillau

This review from Dr. Cambillau’s lab is focused on the baseplate/tail tip complexes from bacteriophages, comparing baseplate proteins structures from phages that infect diverse hosts, using either proteins or carbohydrates as receptors. The main conclusions are that all phages contain a conserved set of core structures, but differ in their receptor binding proteins, as expected for phages that infect different hosts.

This is a timely review, in light of many recent reports on baseplate structures, and certainly appropriate for a Michael Rossmann memorial issue, given that Dr. Rossmann’s collaboration with Dr. Mezyanzhinov on T4 was one of the pioneering high-resolution structural studies of baseplate structures.

The review is quite comprehensive in its scope and well written. The figures are nice, although it would be preferable to put them all on white backgrounds rather than the black backgrounds in several figures presently.

My main criticism of the paper is that it is rather descriptive, and could do with more rigorous and quantitative comparisons in many places. In some cases, the comparisons seem a bit of a stretch in terms of structural similarity, e.g. that of gpJ with Tal. It would be nice to see superpositions of the various domains that are claimed to be similar, with numerical values for their similarity (e.g. RMSD between equivalent Calphas) listed. A table of comparisons that include the HHpred hits, the residues matched, the E-value as well as probability (although the two are presumably highly correlated), and the sequence identity would be helpful. Many of the comparisons made are difficult to evaluate without this information. Additionally, with this information in a table form, some of the text could also be shortened, as the paper is somewhat lengthy at present.

In many cases, the PDB ID is not listed (e.g. for Neisseria MC58 Tal) making it difficult to evaluate the structures.

The paper also contains some inaccuracies throughout that need to be corrected, as listed in more detail below.

Specific comments:

Line 61: It is in general not justified to call E. coli “harmless”, as it includes many serious pathogens. Maybe model strains E. coli K12, B, C and W could be specified.

Line 74: Some bacteria may bind to the non-carbohydrate portion of WTA as well.

Figure 1: A white background would be better.

Line 84: infectaing -> infecting.

Line 94: when genes are mentioned (tmp, dit, tal) they should be italicized, when their gene products are described they should be capitalized and not in italics.

Figure 2: As described in more detail below, the implication from this figure that lambda gpJ and T5 pb3 are homologous with Tal proteins from p2, TP901-1, 80a seems tenuous at best and is not supported by the comparisons. Strictly in functional terms that may be correct, but the proteins seem entirely different.

Line 112: codes -> coded

Line 124 and throughout: Greek letters (alpha and beta) have been lost.

Line 126: “a C-terminal domain” should be plural: “C-terminal domains”

Line 147: It has not been demonstrated experimentally that the 80alpha Tal has enzymatic activity against peptidoglycan or anything else, although it is suggested from sequence prediction. Its Tal CTD is completely different from those of TP901-1 and Tuc2009.

Line 188: Eventual is probably not the right word here. Maybe “possibly” or just “any” would work better.

Line 195-198: The 80a RBP is not an X-ray structure.

Line 198: Use the Greek phi for phi11 here and throughout.

Line 252-254: While it is probably true that gpM corresponds to Dit, it is odd that the HHpred hit is only with the first half of the protein, raising some questions about its validity. A superposition of the two structures, and its corresponding RMSD value might be helpful. Also list HHpred matches in a table (with E value).

Line 264: something is missing before “motif”—alpha/beta maybe?

Line 268-270: Unclear what protein is being referred to here. Please give the PDB ID of the Neisseria MC58 Tal protein. The MC58 Tal protein is not illustrated anywhere in the paper.

Furthermore, the matched segment of MC58 Tal is in the N-terminal domain, not the C-terminal domain, as stated. Most of gpJ doesn’t match any known structures and does not have an NTD like the other examples given, so clearly gpJ is quite different, and one might need to be cautious in drawing too broad conclusions from this comparison.

Line 329-331: The same issue as above regarding the comparison between pb3 and MC58 Tal, but now ~380 residues later in the sequence. Both gpJ and pb3 lack the MTP-like domain that interacts with Dit in other phages. I don't think you can conclude from these comparisons that gpJ and pb3 are homologous with the Tal proteins NTDs from other phages, as suggested by Figure 2. (See also comment above)

Line 335-336: gp3 and gp4 should be “pb3” and “pb4”

Line 351-2: Please list HHpred matches in a table and include E-values.

Line 478: “By similarity with phage TP901-1…” is unclear. Maybe “by comparison” or “by analogy?”

Line 541-542: “…an activation mechanism similar to that…” One has to be careful in drawing a comparison between p2 and T4 here. The two processes are clearly completely different and cannot be said to be “similar.” Only the general concept that there IS an activation mechanism can be said to be related. Please rephrase.

Figures 6, 7 and 8: Please change to white background.

Line 597-599: Please present the information in a table form and list E-values as well as probabilities for the Tal and Dit matches with other phages.

Author Response

Reviewer 1

This review from Dr. Cambillau’s lab is focused on the baseplate/tail tip complexes from bacteriophages, comparing baseplate proteins structures from phages that infect diverse hosts, using either proteins or carbohydrates as receptors. The main conclusions are that all phages contain a conserved set of core structures, but differ in their receptor binding proteins, as expected for phages that infect different hosts.

This is a timely review, in light of many recent reports on baseplate structures, and certainly appropriate for a Michael Rossmann memorial issue, given that Dr. Rossmann’s collaboration with Dr. Mezyanzhinov on T4 was one of the pioneering high-resolution structural studies of baseplate structures.

The review is quite comprehensive in its scope and well written. The figures are nice, although it would be preferable to put them all on white backgrounds rather than the black backgrounds in several figures presently.

My main criticism of the paper is that it is rather descriptive, and could do with more rigorous and quantitative comparisons in many places. In some cases, the comparisons seem a bit of a stretch in terms of structural similarity, e.g. that of gpJ with Tal. It would be nice to see superpositions of the various domains that are claimed to be similar, with numerical values for their similarity (e.g. RMSD between equivalent Calphas) listed. A table of comparisons that include the HHpred hits, the residues matched, the E-value as well as probability (although the two are presumably highly correlated), and the sequence identity would be helpful.

Many of the comparisons made are difficult to evaluate without this information. Additionally, with this information in a table form, some of the text could also be shortened, as the paper is somewhat lengthy at present.

Our answer: Table 1 presents now all the HHpred information concerning Dit and Tal of phages lambda, T5 and SPP1.

In many cases, the PDB ID is not listed (e.g. for Neisseria MC58 Tal) making it difficult to evaluate the structures.

Our answer: PDB IDs have been listed in the text or Table 1. We have also uniformized the call “PDB ID” throughout the text.

The paper also contains some inaccuracies throughout that need to be corrected, as listed in more detail below.

Specific comments:

Line 61: It is in general not justified to call E. coli “harmless”, as it includes many serious pathogens. Maybe model strains E. coli K12, B, C and W could be specified.

Our answer: We have amended the text:

p.2, lines 63-64 "as many strains of this bacterium (e.g. K12) are harmless and very easy to cultivate."

Line 74: Some bacteria may bind to the non-carbohydrate portion of WTA as well.

Our answer: We have amended the text:

p.2, lines 75-76 " bind either to proteinaceous receptors or to carbohydrates (and associated motifs) located at the surface of the host cell wall".

Figure 1: A white background would be better.

Our answer: The revised Figure 1 has a white background. We have also changed the color code (green lines instead of yellow lines) for a better rendering, and the legend has been amended accordingly (p3, line 90).

Line 84: infectaing -> infecting.

Our answer: This has been corrected (p3, line 88).

Line 94: when genes are mentioned (tmp, dit, tal) they should be italicized, when their gene products are described they should be capitalized and not in italics.

Our answer: We have amended the text:

p3, lines 98-99 “Within the latter are the tmp, dit and tal triad, and in most siphophages examined to date, the tmp gene is (immediately) followed by the dit and tal genes (Fig. 2a)”.

Figure 2: As described in more detail below, the implication from this figure that lambda gpJ and T5 pb3 are homologous with Tal proteins from p2, TP901-1, 80a seems tenuous at best and is not supported by the comparisons. Strictly in functional terms that may be correct, but the proteins seem entirely different.

Our answer: Although the reviewer points correctly our mistake - the gp27 like domain is not at the N-terminus, but past the N-terminus - we do not agree with the reviewer’s comment. Indeed, several hits with Tal N-termini domains (shown in Table 1) confirm our analysis that gpJ and pb3 are Tals, with a different topology. We would say that the structure is in part conserved, while the functions of the C-termini are probably different.

The depictions of the lambda and T5 Tal-encoding genes in Figure 2 has been amended considering the reviewer's criticism.

Line 112: codes -> coded

Our answer: This has been corrected (p3, line 121).

Line 124 and throughout: Greek letters (alpha and beta) have been lost.

Our answer: We have reintroduced Greek letters throughout the formatted file for revision.

Line 126: “a C-terminal domain” should be plural: “C-terminal domains”

Our answer: This has been corrected (p5, line 137).

Line 147: It has not been demonstrated experimentally that the 80alpha Tal has enzymatic activity against peptidoglycan or anything else, although it is suggested from sequence prediction. Its Tal CTD is completely different from those of TP901-1 and Tuc2009.

Our answer: We agree with the reviewer. We have amended the text accordingly:

  1. 5. lines 156-160 "In phages TP901-1 and Tuc2009 [36], this extension possesses a lysin domain (hence the original name of Tal) that cleaves the peptidoglycan upon infection. Structure prediction of phage 80a Tal C-terminus suggests that it may also possess a peptidoglycan cleavage activity [15]."

Line 188: Eventual is probably not the right word here. Maybe “possibly” or just “any” would work better.

Our answer: This has been corrected (p6, line 204 “This includes the Dit-Tal diad, the RBP and possibly baseplate ancillary proteins.").

Line 195-198: The 80a RBP is not an X-ray structure.

Our answer: This has been corrected (p6, line 211 “Atomic structures of the RBPs of a small number of siphophages are available […]”.)

Line 198: Use the Greek phi for phi11 here and throughout.

Our answer: This has been corrected (lines 214, 228, 236 and 529).

Line 252-254: While it is probably true that gpM corresponds to Dit, it is odd that the HHpred hit is only with the first half of the protein, raising some questions about its validity. A superposition of the two structures, and its corresponding RMSD value might be helpful. Also list HHpred matches in a table (with E value).

Our answer: The structure of gpM is not available. We think that gpM is the Dit, because it covers 70% of the sequence with good values. It is rare that HHpred covers 100% of the sequence. Furthermore, this Dit is different from the T5 one and others as it lacks the peripheral domain.

We have amended the text to make this point clearer:

p8, lines 278-279 “suggesting that its Dit is only composed of the ring domain, and is devoid of any extension, OB-fold in T5 and galectin in other phages (Fig. 2a).”.

Line 264: something is missing before “motif”—alpha/beta maybe?

Our answer: This has been corrected (p9, line 293 “a a-b-a-b motif”).

Line 268-270: Unclear what protein is being referred to here. Please give the PDB ID of the Neisseria MC58 Tal protein. The MC58 Tal protein is not illustrated anywhere in the paper.Furthermore, the matched segment of MC58 Tal is in the N-terminal domain, not the C-terminal domain, as stated. Most of gpJ doesn’t match any known structures and does not have an NTD like the other examples given, so clearly gpJ is quite different, and one might need to be cautious in drawing too broad conclusions from this comparison.Line 329-331: The same issue as above regarding the comparison between pb3 and MC58 Tal, but now ~380 residues later in the sequence. Both gpJ and pb3 lack the MTP-like domain that interacts with Dit in other phages. I don't think you can conclude from these comparisons that gpJ and pb3 are homologous with the Tal proteins NTDs from other phages, as suggested by Figure 2. (See also comment above)

Our answer: We have addressed the reviewer’s remarks above. The fact that a part of the sequence is not retrieved by HHpred does not mean that the domain is absent or totally different. The explanation is that the sequence identity is too small for HHpred to retrieve a structure.

We have amended Figure 2 and the text accordingly, and listed the PDB IDs in Table 1:

p9, lines 297-301 (lambda gpJ) “ HHpred analysis of gpJ reports hits with the N-terminus of phage Tal proteins (Table 1). However, this segment is not located at the N-terminus of gpJ, but starts at residues 326-330. If we assume that it resembles the classical Tal N-terminus domain of other phages, this domain should cover gpJ residues ~185-520.”.

p11, lines 359-365 (T5 pb3) “HHpred analysis of pb3 reports hits with the N-terminus of phage Tal proteins (Table 1) and with fibronectin III repeats (Table 1). As in phage lambda, this segment is not located at the N-terminus of pb3, but starts at residue ~530. If we assume that it resembles the classical Tal N-terminus domain of other phages, this domain should cover pb3 residues ~386-754 (Table 1, Fig. 4b). The T5 central fiber is formed by pb4, likely as a trimer, and maybe also by the C-terminus of pb3.”.

Line 335-336: gp3 and gp4 should be “pb3” and “pb4”

Our answer: This has been corrected (p16, lines 364-365).

Line 351-2: Please list HHpred matches in a table and include E-values.

Our answer: We have included the Table 1 that lists HHpred matches and values (probability, E-value, and sequence identity).

Line 478: “By similarity with phage TP901-1…” is unclear. Maybe “by comparison” or “by analogy?”

Our answer: This has been corrected (p15, lines 518 “By analogy with TP901-1, […]”).

Line 541-542: “…an activation mechanism similar to that…” One has to be careful in drawing a comparison between p2 and T4 here. The two processes are clearly completely different and cannot be said to be “similar.” Only the general concept that there IS an activation mechanism can be said to be related. Please rephrase.

Our answer: This has been rephrased (p16, lines 584-585 “This represents clearly an activation mechanism related to that described in Myoviridae such as phage T4 [6].”).

Figures 6, 7 and 8: Please change to white background.

Our answer: A white background has been put for Figures 1, 6 and 7. Figure 8 has been removed (see reviewer 2’s comment hereafter).

Line 597-599: Please present the information in a table form and list E-values as well as probabilities for the Tal and Dit matches with other phages.

Our answer: We have included the Table 1 that lists HHpred matches and values (probability, E-value, and sequence identity).

Reviewer 2 Report

A few minor edits.

Line 50: “while” changed to “where”.

Line 58: Please describe the acronym “LAB”

Line 103: You mention the MTP rings are already a component of the IC in line 99, but then mention that MTPs are later attached to the IC. Perhaps the authors could add an extra sentence here to help improve the description of phage tail assembly, and perhaps explain (briefly) the tail terminator?

Line 142: “domain resembling those described above,” does this refer to the introduction of Tals, or do you mean to the Dits and their ring structure? This is unclear. Please describe in more detail your comparison.

Line 231: Regarding PSA RBP: Is it not a fusion of an RBP and a BppU (e.g., TP901-1) instead of what is described in the text? For instance (taken from Dunne et al. 2019… “Bioinformatics analyses predicted the Gp15 N terminus (Leu4–Thr181) has structural similarity to the upper baseplate protein (BppU; ORF48) of phage TP901-1 (Veesler et al., 2012), which we propose is used to connect Gp15 to the phage baseplate.” (https://www.sciencedirect.com/science/article/pii/S2211124719312598#bib67)

Line 446: Change RBP to baseplate.

Line 472: refer to the Dits here as “evolved Dits” for consistency.

Line 525: please describe the acronym “negative stain EM (nsEM)” here.

Line 606: It is really not clear to me what the role of TssK is here? Is the role of TssK known? Also I am not entirely sure what the reader is meant to understand from figure 8 and why the connections of the p2 RBPs and T6SS is being investigated in such detail. Are these also crystal structures or outputs of modelling?

Line 622: change “along” to “through”

Authors provide a structure-orientated review of Siphophage baseplate organisations. The authors describe the differences between These baseplates targeting Protein or Saccharidic receptor; however, they don't really explain any reasoning why the baseplates are organised differently for different Targets, e.g., does avidity come into play for sugar-binding baseplates requiring multiple RBP binding sites vs. T5 and lambda targeting Protein receptors?

Author Response

Reviewer 2

Line 50: “while” changed to “where”.

Our answer: This has been corrected (p2, line 52).

Line 58: Please describe the acronym “LAB”

Our answer: We have added the description of “LAB” (p2, lines 59-61 “Phages that infect lactic acid bacteria (LAB phages) involved in fermentation processes are over-represented […]”).

Line 103: You mention the MTP rings are already a component of the IC in line 99, but then mention that MTPs are later attached to the IC. Perhaps the authors could add an extra sentence here to help improve the description of phage tail assembly, and perhaps explain (briefly) the tail terminator?

Our answer: we have amended the text to improve the description of phage tail assembly:

p3, lines 105-109 “All together, TMP, Dit and Tal participate to the first assembly module of the tail, the Initiator Complex (IC) [20, 21]. This IC complex is then completed by the RBP and other baseplate proteins, if present, as well as with the MTPs and the tail terminator, the protein that blocks the tail extension, resulting in the complete tail that is ready to attach to the capsid connector [20, 22].”.

Line 142: “domain resembling those described above,” does this refer to the introduction of Tals, or do you mean to the Dits and their ring structure? This is unclear. Please describe in more detail your comparison.

Our answer: we have amended the text to clarify our comparison:

p5, lines 151-154 “HHpred [33] analysis of these Tals reveals that they possess a domain resembling the Tal domains mentioned above, for which the structures have been determined, followed by an extension exhibiting large structural and functional diversity.”.

Line 231: Regarding PSA RBP: Is it not a fusion of an RBP and a BppU (e.g., TP901-1) instead of what is described in the text? For instance (taken from Dunne et al. 2019… “Bioinformatics analyses predicted the Gp15 N terminus (Leu4–Thr181) has structural similarity to the upper baseplate protein (BppU; ORF48) of phage TP901-1 (Veesler et al., 2012), which we propose is used to connect Gp15 to the phage baseplate.” (https://www.sciencedirect.com/science/article/pii/S2211124719312598#bib67)

Our answer: We agree with the reviewer. We have corrected our mistake:

p8, lines 257-258 “ In listerial phage PSA, the RBP results from the fusion of a RBP module and a BppU module (as those of phage TP901-1), that binds to the rest of the baseplate [13].”

Line 446: Change RBP to baseplate.

Our answer: This has been corrected (p13, line 470-471 “The 18 BppU assemble as six asymmetric trimers connecting the Dit central core.”).

Line 472: refer to the Dits here as “evolved Dits” for consistency.

Our answer: we have amended the text (p15, lines 510-512 “The BppA CBM was later found to be ubiquitous in decorations of lactococcal and streptococcal phages evolved Dits or MTPs [32, 43, 88].”).

Line 525: please describe the acronym “negative stain EM (nsEM)” here.

Our answer: we have amended the text to describe the acronym nsEM:

p16, line 566-569 “However, there was no doubt on the validity of the structure of the baseplates crystal structure as this conformation was found in the negative staining EM (nsEM) structure of the expressed baseplate as well as in the full virion nsEM structure (Fig. 6a).”.

Line 606: It is really not clear to me what the role of TssK is here? Is the role of TssK known? Also I am not entirely sure what the reader is meant to understand from figure 8 and why the connections of the p2 RBPs and T6SS is being investigated in such detail. Are these also crystal structures or outputs of modelling?

Our answer: In order to avoid any misunderstanding, we have removed Figure 8.

Line 622: change “along” to “through”

Our answer: This has been corrected (p19, line 664).

Authors provide a structure-orientated review of Siphophage baseplate organisations. The authors describe the differences between These baseplates targeting Protein or Saccharidic receptor; however, they don't really explain any reasoning why the baseplates are organised differently for different Targets, e.g., does avidity come into play for sugar-binding baseplates requiring multiple RBP binding sites vs. T5 and lambda targeting Protein receptors?

Our answer: we have amended the introduction to section 4:

p13, lines 455-460 “Whereas phages that bind to a host protein have an elongated tail tip, phages binding to cell wall surface polysaccharides possess a bulky distal tail feature named the baseplate. This large structure encompasses a high number of RBPs (typically 18 to 54) as avidity is required to overcome the rather low RBP affinity for sugars, as compared to high protein-protein affinity observed for example in phage T5 [71]. The highlighted examples below illustrate the conserved and diverse parts of these adhesion devices.”.
